# Hydrophobic Fluorinated Porous Organic Frameworks for Enhanced Adsorption of Nerve Agents

**Shuyuan Zhou [1,\*], Weimin Kong [1], Kunpeng Wang [1], Hao Han [1], Derui Yang [1], Yue Zhao [1], Hong Zhou [1], Qinghao Meng [2] and Ye Yuan [2,\*]**

[1] State Key Laboratory of NBC Protection for Civilian, Research Institute of Chemical Defense, Academy of Military Sciences, Beijing 102205, China; kongweim123@hrbeu.edu.cn (W.K.); email151958@163.com (K.W.); zhangc466@nenu.edu.cn (H.H.); fengjh010@nenu.edu.cn (D.Y.); SA11226532@mail.ustc.edu.cn (Y.Z.); zhouhongbuct@126.com (H.Z.)

[2] Key Laboratory of Polyoxometalate Science of Ministry of Education, Northeast Normal University, Changchun 130024, China; Mengqh954@nenu.edu.cn

\* Correspondence: zhoushuyuan1987@126.com or wangxw5710@hrbeu.edu.cn (S.Z.); Yuany101@nenu.edu.cn (Y.Y.)



**Featured Application: Capture of chemical warfare agents.**

**Abstract:** Humidity in the air can significantly limit the adsorption capacity of porous materials used for the removal of chemical warfare agents (CWAs). Therefore, in this work, we prepared a porous organic material (C-1) and its fluoride derivative (C-1-F) via a Schiff base reaction and determined their structure and morphological properties, hydrophobicity, and adsorption capacity. Compared to the parent C-1 material, both the channel and particle surface of C-1-F were highly hydrophobic, thus stabilizing the fluorinated porous material under various humidity conditions. Dimethyl methyl phosphonate was used as a nerve agent simulant to examine the efficiency of the synthesized porous materials, indicating that C-1-F had a higher adsorption capacity than C-1 under dry conditions. Moreover, unlike C-1, the adsorption capacity of hydrophobic C-1-F was not affected even under a relative humidity of 20%, and it is still able to maintain high adsorption capacity at a relative humidity of 60%, suggesting its high application potential in the removal of CWAs.

**Keywords:** chemical warfare agents; nerve agents; porous organic frameworks; hydrophobic surface; adsorption

## 1. Introduction

Chemical warfare agents (CWAs) have been widely used in World Wars I and II, leading to massive human casualties due to their strong toxicity, rapid action, and low lethal inhalation dose. CWAs are extremely poisonous to humans and animals upon release [1], and the destruction of remaining stocks from the war remains a global concern. Although CWAs have been banned by the Chemical Weapons Convention, they can be easily obtained by terrorist organizations, posing a global threat. Therefore, the development of high-performance adsorbents for the effective capture of CWAs is urgently needed.

Porous materials have been extensively studied for the adsorption and catalysis of CWAs [2–12]. Activated and impregnated activated carbons (such as metal salts, acids, and organic amines) are the most commonly used materials for CWA capture [13–15], while metal–organic frameworks (MOFs) and zeolites have also attracted significant attention in recent years [16–20]. However, water molecules from atmospheric humidity strongly compete with CWAs for the porous materials' adsorption sites,

thus significantly limiting their adsorption capacity. Therefore, several hydrophobic MOFs have recently been developed which could effectively capture sarin $[[(CH_3)_2CHO]CH_3P(O)F]$ and mustard gas $[(ClCH_2CH_2)_2S]$ simulants, even in the presence of mimic ambient moisture [21,22].

Porous organic frameworks (POFs) have also been widely used in various fields, such as gas storage and separation, heterogeneous catalysis, energy storage, etc., as they combine the properties of porous materials and polymers [23–36]. It is thus expected that POFs would effectively adsorb CWAs due their large surface area, tunable pore size, and functionable channel. However, their use for the removal of CWAs has not yet been reported, indicating that additional hydrophobic porous materials with high adsorption capacity need to be explored.

In this study, a new conventional POF material, denoted as C-1, was synthesized using 1,3,5-tris(4-aminophenyl)benzene and 1,4-phthalaldehyde. To increase the hydrophobicity of C-1, 2,3,5,6-tetrafluoroterephthaldehyde, containing the hydrophobic –F group, was used as fluorinated linker, resulting in a novel POF material, named C-1-F. The surface and channel hydrophobicity of C-1 and C-1-F were characterized by contact angle and water vapor adsorption isotherms, while dimethyl methyl phosphonate (DMMP) was used as a safe simulant of the nerve agent sarin due to their similar molecular structures and volatilities. The results showed that the fluorinated POF had a significantly enhanced adsorption capacity for DMMP compared to C-1 under both dry and wet air conditions.

## 2. Materials and Methods

### 2.1. Materials

1,4-Phthalaldehyde, 1,3,5-tris(4-aminophenyl)benzene, dioxane, and 1,3,5-trimethylbenzene were purchased from Ailan Chemical Technology Co., LTD (Shanghai). Acetic acid, acetone, N,N-dimethyl-formamide (DMF), and DMMP were purchased from Shanghai Macklin Biochemical Technology Co., LTD. All chemicals were used as received. Activated carbon were purchased from Xingchang Active Carbon Co., LTD (Jiangsu). Activated carbon was dried at 120 °C for 12 h before using.

### 2.2. Characterization of C-1 and C-1-F

The powder X-ray diffraction (XRD) measurements were performed on a SmartLab instrument with Cu K$\alpha$ ($\lambda$ = 1.5418 Å) radiation at 40 kV and 30 mA over an angular range of $2\theta$ = 4–40° at a scan rate of 10 °C·min$^{-1}$. The scanning electron microscopy (SEM) images were recorded on a JSM 6700 and a Hitachi SU8000 scanning electron microscope at 3 kV and 5 μA. The sample (1 mg) was dispersed in 2 mL ethanol using ultrasound, and the dispersed droplets were placed on a clean silicon wafer. After ethanol evaporation, the wafer-supported sample was kept at 60 °C for 12 h and then tested by SEM. The Fourier transform infrared (FT–IR) spectra were recorded on a Nicolet Impact 410 FT–IR spectrometer. The $N_2$ adsorption isotherms and the pore size distribution were determined using a Micromeritics ASAP 2010M analyzer. Water vapor adsorption isotherms were measured at 298 K using a Beishide vacuum steam adsorption apparatus. Thermogravimetric analysis (TGA) was performed on a Mettler Toledo TGA/DSC 3+ analyzer at a heating rate of 10 °C·min$^{-1}$ under air flow.

### 2.3. Synthesis of C-1 and C-1-F

1,3,5-Tris(4-aminophenyl)benzene (55 mg, 0.16 mmol), 1,4-benzenedialdehyde (31 mg, 0.23 mmol), and a mesitylene (5.94 mL)/dioxane (1.26 mL) mixture were added to a 10 mL Pyrex tube. After sonication for 20 min, acetic acid (1.2 mL) and distilled water (1.8 mL) were added to the reaction and the mixture was degassed under liquid $N_2$ (77 K) by applying three freeze–pump–thaw cycles using a Schlenk line (with $N_2$). The Pyrex tube was then sealed under static vacuum using an alcohol torch and kept at 120 °C for 3 days. The formed solid was washed with DMF and acetone and dried at 60 °C overnight to afford C-1 in 94% isolated yield. C-1-F was obtained in ~95% isolated yield following the same procedure and using 2,3,5,6-tetrafluoroterephthalaldehyde instead of 1,4-benzenedialdehyde. The schematic diagram of synthetic route is shown in Figure 1.

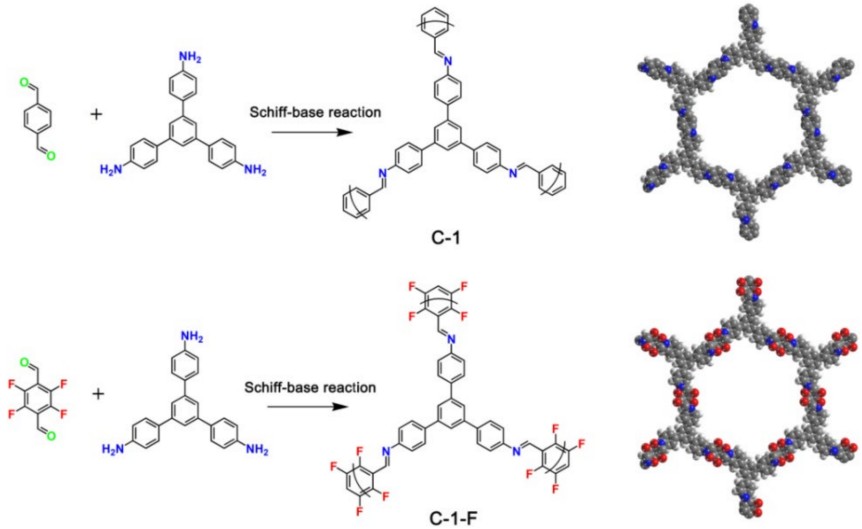

**Figure 1.** The schematic diagram of synthetic route.

*2.4. DMMP Dynamic Adsorption Performance Measurements*

A dynamic adsorption experimental equipment was established to evaluate the adsorption performance of DMMP (Figure 2). In particular, a constant flux of dry air (190 mL·min$^{-1}$) was bubbled through DMMP contained in a flask at 32 °C. DMMP was then swept with dry air (630 mL·min$^{-1}$) in a gas mixing chamber. For experiments with different relative humidity, dry air (630 mL·min$^{-1}$) was bubbled in a flask containing distilled water at room temperature equipped with a thermohygrograph recording the relative humidity. Samples (9–12 mg) were then loaded into a tray suspended by a quartz spring. The adsorption of DMMP was evaluated by monitoring the changes in the length of the quartz spring. Based on Hooke's law, the spring elongation is proportional to the mass of the hanging object. Thus, the dynamic adsorption capacity was calculated based on the relative weight (length) ($M = L$) increase as follows:

$$M = L = \frac{L_{\text{sample}} - L_0}{L_{\text{saturation}} - L_{\text{sample}}} \times 100\% \tag{1}$$

where $L_0$, $L_{\text{sample}}$, and $L_{\text{saturation}}$ are the stretched length of the quartz spring for pure tray, sample-loaded tray, and saturated absorption tray, respectively.

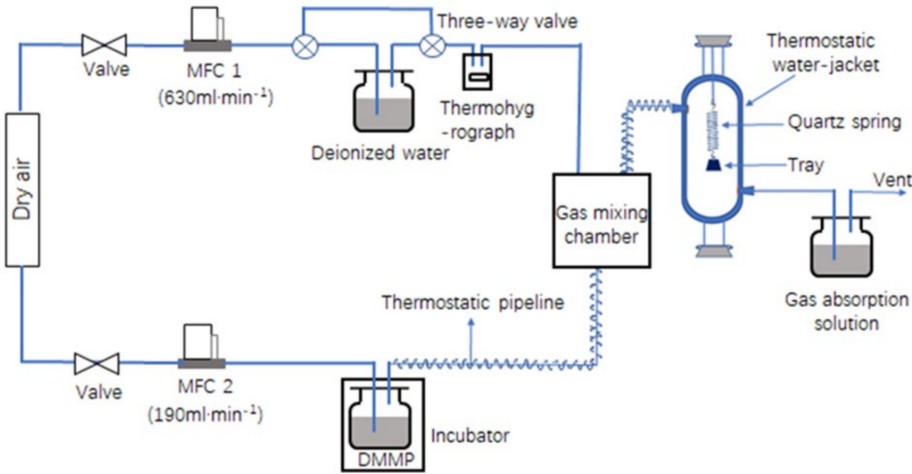

**Figure 2.** The schematic of the dynamic adsorption experimental setup.

## 3. Results and Discussion

### 3.1. Structure and Morphology

FT–IR spectroscopy was used to confirm the successful preparation of C-1 and C-1-F. As shown in Figure 3, the characteristic peaks of the aldehyde and amino groups at 1700 and 3300–3500 cm$^{-1}$, respectively, disappeared, indicating that both materials were successfully synthesized.

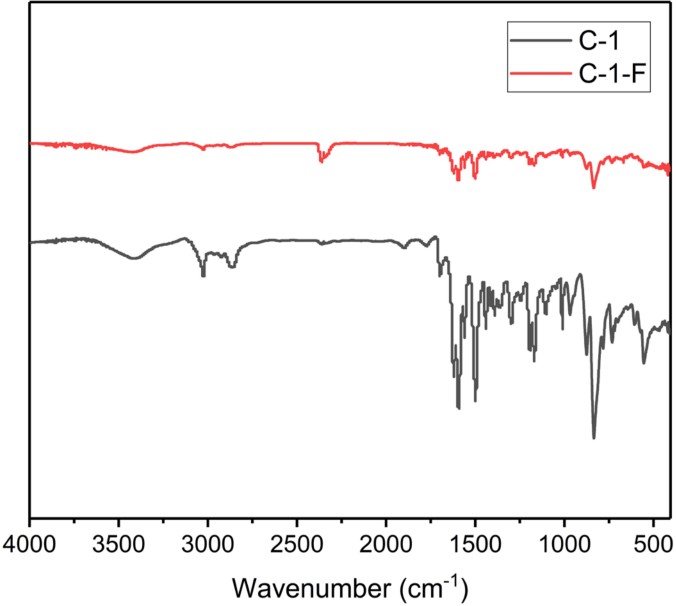

**Figure 3.** FT–IR spectra of C-1 and C-1-F.

Moreover, XRD studies on the crystallinity of the synthesized materials (Figure 4) showed that no characteristic diffraction peaks were observed between 4–40°, indicating that both C-1 and C-1-F were amorphous materials. SEM measurements (Figure 5) also revealed that C-1 and C-1-F had quite different morphologies. C-1 had a hollow network structure, while C-1-F formed irregular hollowed-out spheres consisting of 100–200 nm elliptic particles.

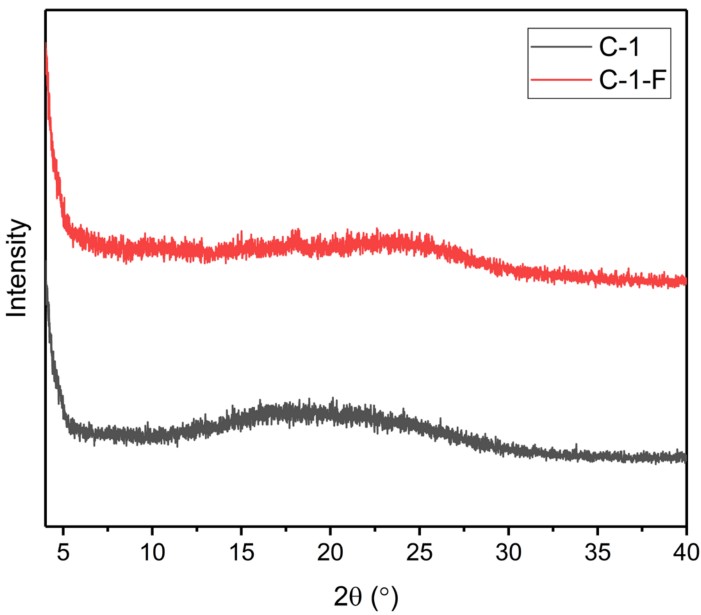

**Figure 4.** XRD diffraction patterns of C-1 and C-1-F.

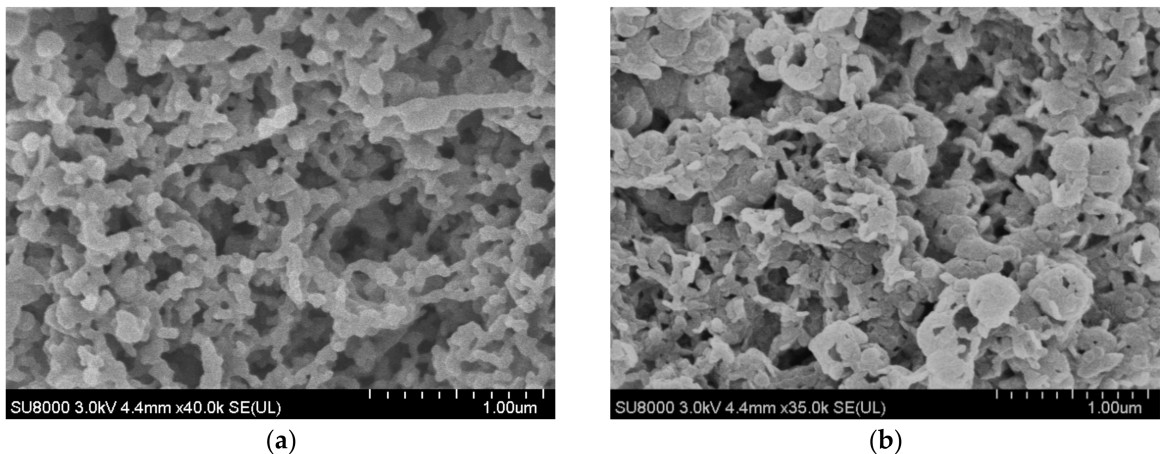

**Figure 5.** SEM pictures of (**a**) C-1 and (**b**) C-1-F.

The obvious hysteresis loop observed on the $N_2$ adsorption isotherms at 77 K indicated that the existence of mesopores in both materials (Figure 6a,b) with a Brunauer–Emmett–Teller (BET) surface area of 344 and 434 $m^2 \cdot g^{-1}$ for C-1 and C-1-F, respectively. Based on further density functional theory studies (DFT), the pore size was also widely distributed around 1.1 and 1.4 nm, respectively (Figure 6c,d).

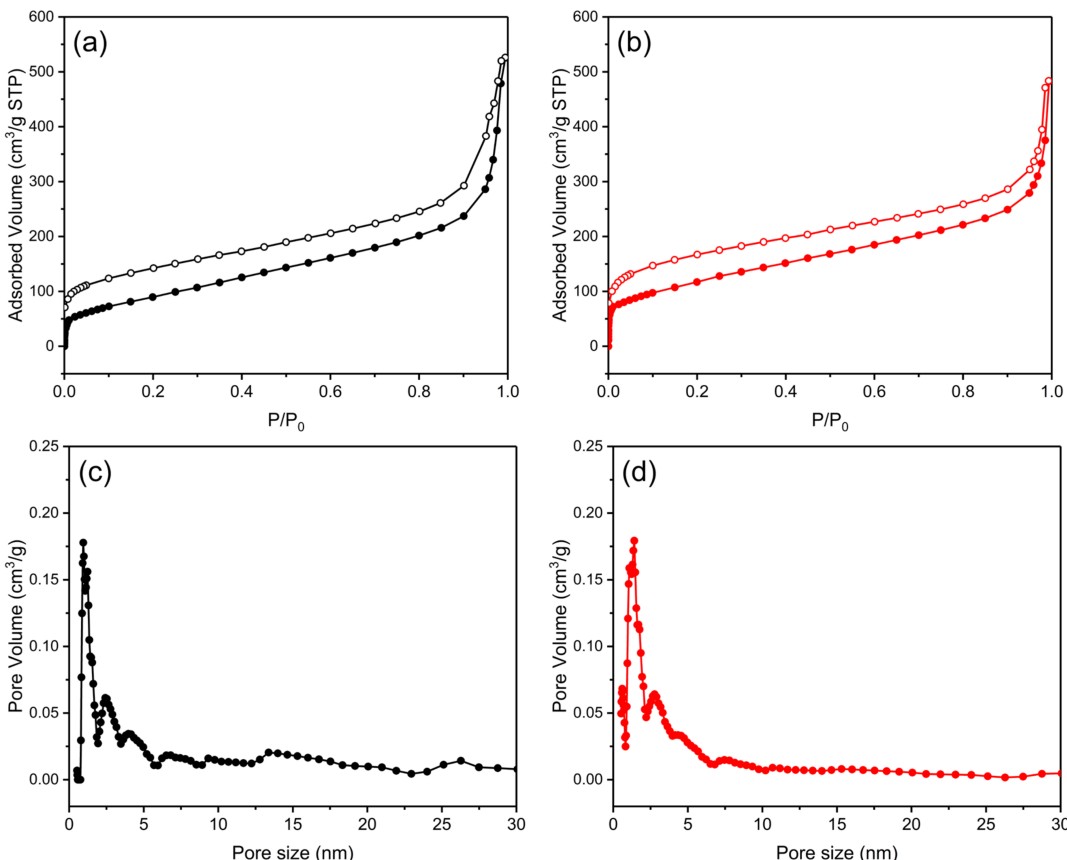

**Figure 6.** $N_2$ adsorption isotherms of (**a**) C-1 and (**b**) C-1-F at 77 K. Pore size distribution curve of (**c**) C-1 and (**d**) C-1-F. Solid and open symbols denote adsorption and desorption, respectively.

### 3.2. Stability

The thermal stability of C-1 and C-1-F was tested by TGA. As shown in Figure 7, both materials had excellent thermal stability and almost no weight loss was observed at temperatures below 400 °C. However, with increasing temperature (>400 °C), a gradual weight loss was observed, and the porous skeleton collapsed at about 650 °C.

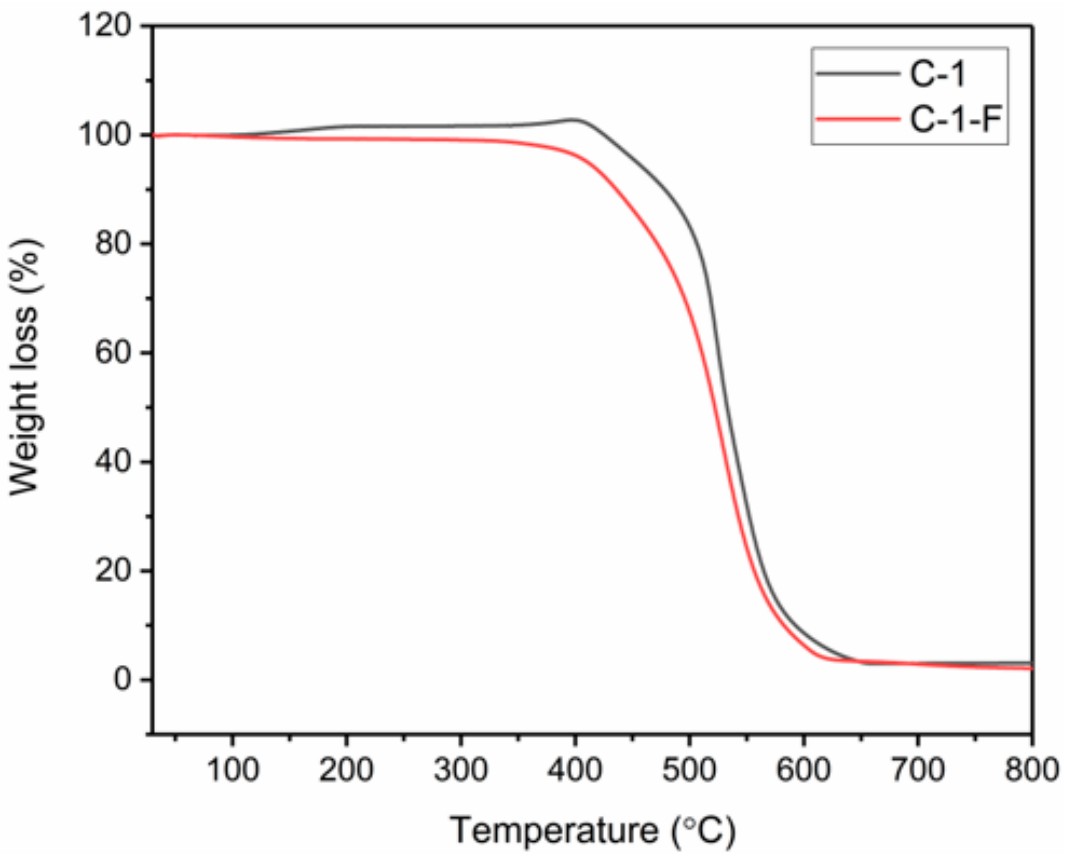

**Figure 7.** TGA curves of C-1 and C-1-F.

To further investigate the effect of humidity on the stability of C-1 and C-1-F, both materials were kept in a saturated standard salt solution environment with controlled relative humidity of 33%, 53%, 65%, 75%, 85%, and 97% for 24 h. As the relative humidity increased, the intensity of the characteristic IR peaks of the aldehyde and amino groups of C-1 at 1690 and 3300–3500 $cm^{-1}$, respectively, gradually increased, indicating the decomposition of the C-1 skeleton (Figure 8a). By contrast, the characteristic peaks of C-1-F remained almost unchanged (Figure 8b), suggesting that the stability of C-1-F was improved by fluorine doping.

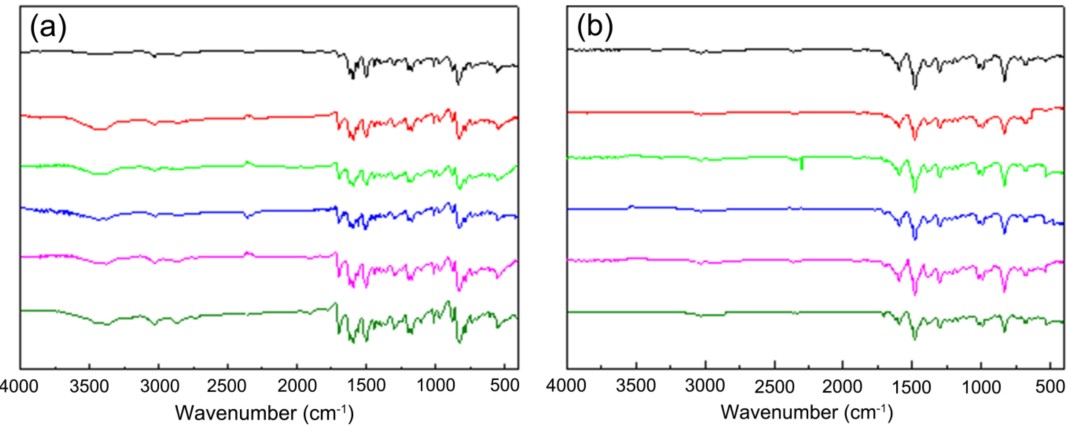

**Figure 8.** FT−IR spectra of (**a**) C-1 and (**b**) C-1-F under different relative humidity conditions. (Black, 33% relative humidity; red, 53% relative humidity; fluorescent green, 65% relative humidity; blue, 75% relative humidity; purple, 85% relative humidity; dark green, 97% relative humidity.)

### 3.3. Surface and Pore Hydrophobicity

The surface and pore hydrophobicity of the synthesized materials were examined by contact angle and water vapor adsorption tests. The contact angles of C-1 and C-1-F were determined at 74 and 98°, respectively (Figure 9). Given that the boundary between hydrophilicity and hydrophobicity is 90°, it is clear that C-1 changed from a hydrophilic to a hydrophobic wetting behavior after fluorine doping.

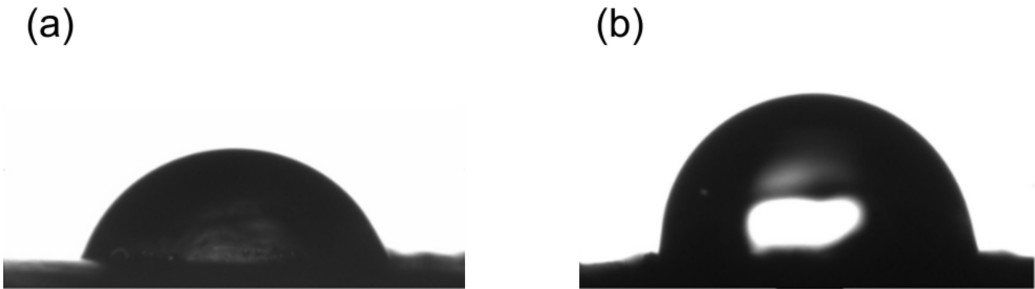

**Figure 9.** Contact angles determined for (**a**) C-1 and (**b**) C-1-F.

Furthermore, the hysteresis loop area observed for C-1-F was smaller than that for C-1 (Figure 10), suggesting that the adsorbed water vapors could be easily desorbed from the material. This narrowing hysteresis loop also suggested a weaker interaction between C-1-F and water vapor due to the more hydrophobic C-1-F channel.

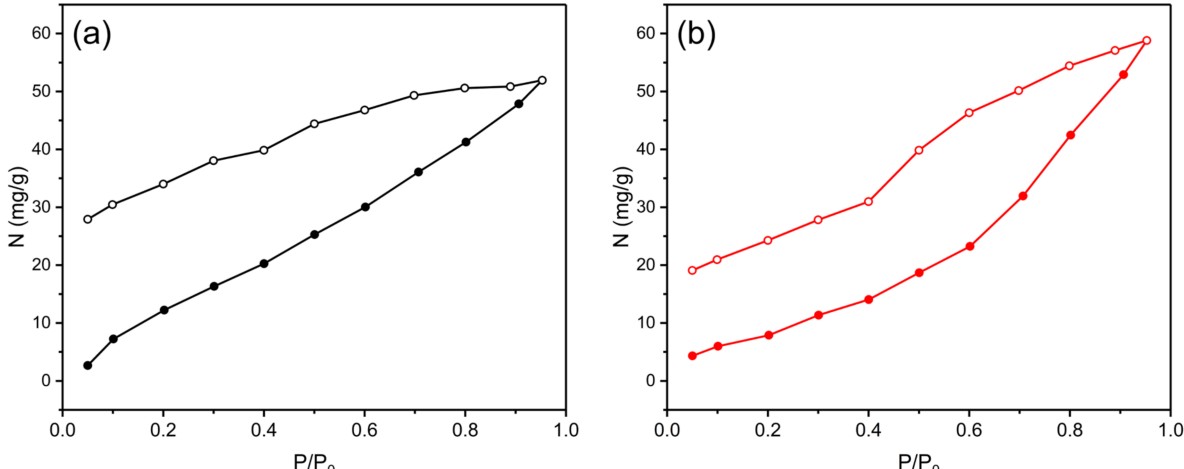

**Figure 10.** Water vapor physisorption isotherms (298 K) obtained for (**a**) C-1 and (**b**) C-1-F. Solid and open symbols denote adsorption and desorption, respectively.

### 3.4. Dynamic Adsorption of DMMP

The dynamic adsorption of DMMP was evaluated under air conditions with a relative humidity of 0%, 20%, and 60%, respectively. To conduct DMMP adsorption experiment under a certain humidity that is comparable to those between C-1 and C-1-F, 20% relative humidity was used whereby C-1 remained stable. In addition, the water vapor dynamic adsorption experiment was carried out at a 20% relative humidity with similar conditions as for the DMMP adsorption experiment. According to Figure 11, characteristic peaks of FT–IR spectra remain relatively unchanged after the water vapor dynamic adsorption, which verifies the stable existence of C-1 at 20% relative humidity condition. Furthermore, since C-1-F exhibits perfect stability even under high relative humidity (Figure 8b), we have also performed the dynamic adsorption of DMMP under a relative humidity of 60%, which presents a closer approximation to the real operating environment.

The adsorption capacities of C-1 and C-1-F for DMMP and the effect of humidity on their performance were compared with the corresponding results obtained for activated carbon. Under dry air, both C-1 and C-1-F showed a dynamic adsorption performance for DMMP and the adsorption capacity of C-1-F (13.73%) was higher than that of C-1 (8.15%) and lower than that of activated carbon (22.39%) (Figure 12a and Table 1).

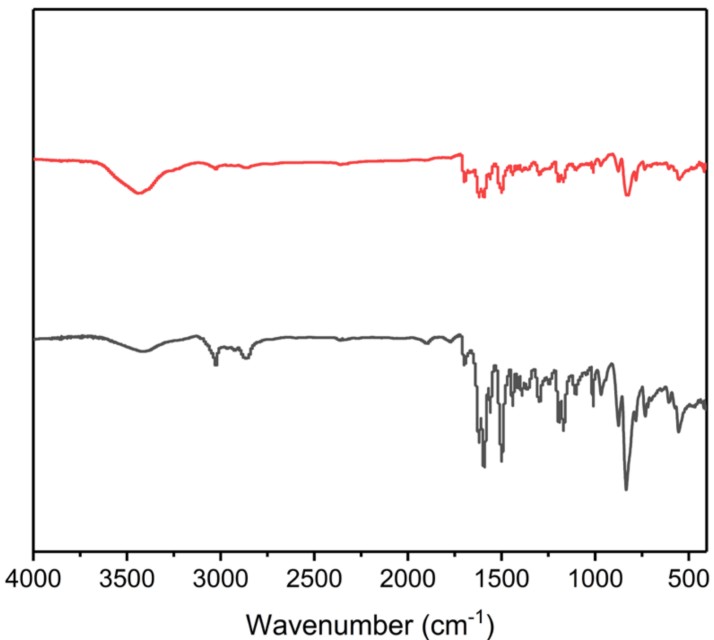

**Figure 11.** FT−IR spectra of C-1 (black) and C-1 after water vapor dynamic adsorption under 20% relative humidity conditions (red).

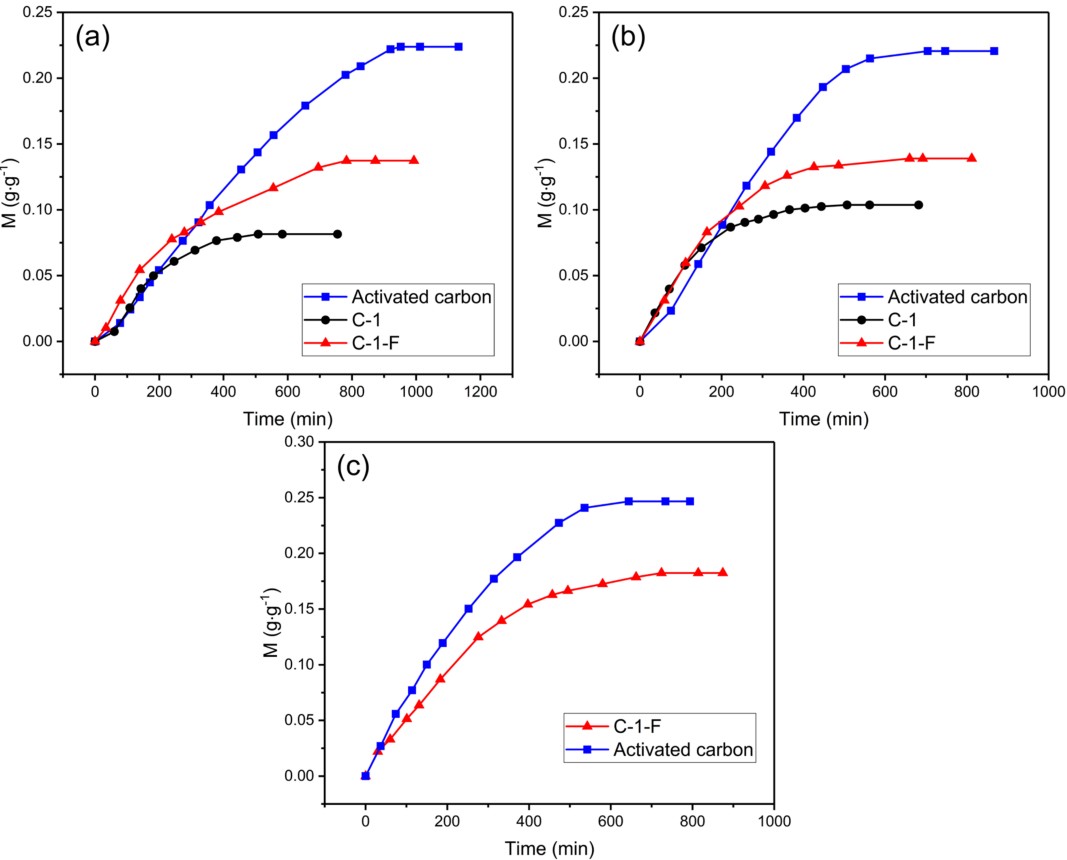

**Figure 12.** Dynamic adsorption capacity of C-1, C-1-F, and activated carbon for DMMP at 305 K under (**a**) dry, (**b**) 20% relative humidity and (**c**) 60% relative humidity conditions. M: relative weight increase.

**Table 1.** BET surface area, pore volume, and relative weight increase of C-1, C-1-F, and activated carbon.

| | Surface area $(cm^2 \cdot g^{-1})$ | $V_{total}$ [1] $(cm^3 \cdot g^{-1})$ | $V_{micro}$ [2] $(cm^3 \cdot g^{-1})$ | $V_{meso}$ [3] $(cm^3 \cdot g^{-1})$ | $M_{0\%}$ [4] | $M_{20\%}$ [5] | $M_{60\%}$ [6] |
|---|---|---|---|---|---|---|---|
| C-1 | 344 | 0.63 | 0.12 | 0.51 | 8.15% | 10.37% | / |
| C-1-F | 434 | 0.50 | 0.16 | 0.34 | 13.73% | 13.90% | 18.24% |
| Activated carbon | 558 | 0.31 | 0.23 | 0.08 | 22.39% | 22.06% | 24.10% |

[1] Total pore volume, [2] micropore volume, [3] mesopore volume, [4,5,6] relative weight increase under relative humidity conditions of 0%, 20%, and 60%.

The adsorption capacity was further analyzed based on the BET specific surface area and pore volume of the synthesized materials. As shown in Table 1, C-1 had the highest total pore volume but the lowest BET surface area, indicating its higher mesoporosity compared to C-1-F. These differences could be attributed to the –F group in C-1-F, which was introduced into the mesoporous channel of C-1, thus forming more micropores and increasing the specific surface area of C-1-F. Based also on the law of volume change between micropores and mesopores, the adsorption capacity of DMMP was positively related with the micropore volume of C-1-F, further confirming its higher DMMP adsorption capacity compared to C-1. Moreover, the strongly electronegative nature of F can result in the formation of electrostatic repulsive interactions between the F atoms with F atoms or any negatively charged species [37–40], which may be a factor for its high adsorption capacity for C-1-F.

The effect of water vapor on the adsorptive performance of materials was also explored (Figure 12b). For increasing the relative humidity to 20%, C-1-F and activated carbon has nearly unchanged adsorption capacity as that under dried conditions (Table 1). Nonetheless, C-1 with a 2.22% adsorption capacity changing indicates the water vapor does affect the adsorption capacity of C-1 materials. As the relative humidity increases to 60%, the effect of water vapor on the adsorption performance of C-1-F and activated carbon increases. However, the sample is still able to exhibit high adsorption capacity (Figure 12c and Table 1).

## 4. Conclusions

In this study, we report for the first time the application of POFs in capturing CWAs. Between the two synthesized POF materials, the fluorinated porous solid exhibited outstanding performance for the adsorption of DMMP in both dry and humid environment due to its enhanced hydrophobicity, large surface area, functionable channel and, possibly, the strongly electronegative nature of fluorine. The high adsorption performance and enhanced stability of C-1-F at high relative humidity leads to POF materials with huge potential for practical application in CWA removal.

**Author Contributions:** Conceptualization, S.Z. and Y.Y.; writing (original draft preparation), S.Z.; writing (review and editing), S.Z. and Y.Y.; supervision, S.Z. and H.H.; Investigation, W.K. and Q.M.; visualization, K.W.; resources, D.Y. and Y.Z.; project administration, H.Z. All authors have read and agreed to the published version of the manuscript.

**Funding:** This research was funded by National Natural Science Foundation of China (21701186).

**Acknowledgments:** We gratefully acknowledge the support from National Natural Science Foundation of China (21701186).

**Conflicts of Interest:** The authors declare no conflict of interest.

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
