# Peer review of "Hydrophobic Fluorinated Porous Organic Frameworks for Enhanced Adsorption of Nerve Agents"

_applsci, doi:10.3390/app10248789_

Round 1
Reviewer 1 Report
Shuyuan Zhou and Ye Yuan et al. reported the synthesis and nerve agent adsorption property of a novel hydrophobic fluorinated porous organic material. The adsorption of DMMP as nerve agent simulants using the POF under wet condition shows unchanged adsorption capacity as that under dried conditions, suggesting its high application potential in the removal of chemical warfare agents. The manuscript is well written and can be accepted to publication after minor revision.
- Why did you perform the POF DMMP adsorption experiments at 20% relative humidity? The authors should state why.
- There are some typos, please correct them.
i.e.
L 20. --nerve agent stimulant – to nerve agent simulant.
L 125. Figures6c and d to Figures 6c and d
L 141 FT-IR spectras to FT-IR spectra
L 143. Dark green to dark green
- Please check again carefully regarding references. Some typos errors are found.
i.e.
L 201, “r-PbO2”
L 207, “modifified” and “metalorganic”
L 247, “Angew. Chem. Int. Ed. Engl”, Engl is unnecessary.
L 259, “metalorganic”
Reviewer 2 Report
This is an interesting innovative work. The fluorine effect discovered by the authors may have very important practical applications. However, the authors conclusion about "hydrophobicity" as the only factor responsible for the dramatic effect is incomplete. In my opinion, while the hydrophobicity plays some important role, the electrostatic repulsive interactions between F atoms as well as between F and anything negatively charged ought to play the major role. The authors should consider this possibility in the discussion. The authors can find examples of pronounced electrostatic repulsive interactions effect on synthetic and physicochemical properties of organic compounds in the following publications from my group: 1) Soloshonok, V. A.; Hayashi, T. Gold(I)-Catalyzed Asymmetric Aldol Reaction of Fluorinated Benzaldehydes with a-Isocyanoacetamide, Tetrahedron: Asymmetry 1994, 5, 1091-1094; 2) Soloshonok, V. A.; Kirilenko, A. G.; Galushko, S. V.; Kukhar, V. P. Catalytic Asymmetric Synthesis of b-Fluoroalkyl-b-Amino Acids via Biomimetic [1,3]-Proton Shift Reaction, Tetrahedron Letters 1994, 35, 5063-5064. 3) A. E. Sorochinsky, T. Katagiri, T. Ono, A. Wzorek, J. L. Aceña, V. A. Soloshonok, Optical purifications via Self-Disproportionation of Enantiomers by achiral chromatography; Case study of a series of α-CF3-containing secondary alcohols, Chirality 2013, 25, 365–368; 4) Soloshonok, V. A.; Avilov, D. V.; Kukhar, V. P. Asymmetric Aldol Reactions of Trifluoromethyl Ketones with a Chiral Ni(II) Complex of Glycine: Stereocontrolling Effect of the Trifluoromethyl Group, Tetrahedron 1996, 52, 12433-12442.
Author Response
Please see the attachment.

This manuscript is a resubmission of an earlier submission. The following is a list of the peer review reports and author responses from that submission.